# Intermittent Fasting, Dietary Modifications, and Exercise for the Control of Gestational Diabetes and Maternal Mood Dysregulation: A Review and a Case Report

**DOI:** 10.3390/ijerph17249379

**Published:** 2020-12-15

**Authors:** Amira Mohammed Ali, Hiroshi Kunugi

**Affiliations:** 1Department of Mental Disorder Research, National Institute of Neuroscience, National Center of Neurology and Psychiatry, Tokyo 187-0031, Japan; hkunugi@ncnp.go.jp; 2Department of Psychiatric Nursing and Mental Health, Faculty of Nursing, Alexandria University, Alexandria 21527, Egypt; 3Department of Psychiatry, Teikyo University School of Medicine, Tokyo 173-8605, Japan

**Keywords:** gestational diabetes mellitus, Egypt, depression, distress, anxiety, exercise, caloric restriction, intermittent fasting, dietary restriction, low glycemic index diet, gut microbiome, insulin resistance, placental hormones, insulin estrogenization, high protein diet, poverty, obesity, fermented milk/yoghurt, soy

## Abstract

Gestational diabetes mellitus (GDM) is a common pregnancy-related condition afflicting 5–36% of pregnancies. It is associated with many morbid maternal and fetal outcomes. Mood dysregulations (MDs, e.g., depression, distress, and anxiety) are common among women with GDM, and they exacerbate its prognosis and hinder its treatment. Hence, in addition to early detection and proper management of GDM, treating the associated MDs is crucial. Maternal hyperglycemia and MDs result from a complex network of genetic, behavioral, and environmental factors. This review briefly explores mechanisms that underlie GDM and prenatal MDs. It also describes the effect of exercise, dietary modification, and intermittent fasting (IF) on metabolic and affective dysfunctions exemplified by a case report. In this patient, interventions such as IF considerably reduced maternal body weight, plasma glucose, and psychological distress without any adverse effects. Thus, IF is one measure that can control GDM and maternal MDs; however, more investigations are warranted.

## 1. Introduction

Gestational diabetes mellitus (GDM) is an abnormal metabolic condition characterized by hyperglycemia during pregnancy [1,2]. GDM impacts around 5% of pregnant women in the United States [3], and reports from other countries indicate that it may affect more than 36% of pregnant women [4,5]. The prevalence of GDM is associated with increased obesity and type 2 diabetes mellitus (T2DM) resulting from an unhealthy diet and sedentary lifestyle [6,7]. It is also associated with poverty and pre-pregnancy obesity [8,9], sleep apnea, poor sleep quality, and short sleep [8]. These factors may increase the odds of developing GDM [9].

In Egypt, in addition to poverty, obesity and malnutrition are highly prevalent, particularly in the young [10]. Obesity among Egyptian females strongly correlates with prevalence of GDM, greater expression of diabetes-related autoantibodies, and subsequent development of T2DM after childbirth [11,12]. Other risk factors for GDM in Egypt include maternal age over 35 years, high blood pressure and pre-eclampsia, family history of diabetes, urban living, lower levels of education, and prior history of GDM, abortion, induced labor, and macrosomia (large for gestational age) [13,14,15]. Unfortunately, reports on the prevalence of GDM in Egypt (6–8.9%) originate from a few small-scale studies [13,14,15]. Nevertheless, this prevalence is considerably lower than the overall stated prevalence of GDM in Africa (13.6%) [16], and it is less likely to reflect the actual magnitude of GDM in Egypt given that the studies may not be representative of the population—the sample sizes were relatively small (130–700 participants), and estimation of a sample size was not performed in any of these studies. Furthermore, the subjects included were from small cities or rural areas. In these regions, women do not usually have proper access to healthcare services (e.g., due to lack of transportation and lower levels of education): urban residency and lower levels of education affect the prevalence of GDM in Egypt [13,14,15].

GDM causes numerous short- and long-term complications, both in the mother and in her offspring [17]. It predisposes the pregnant women to hypertension and preeclampsia [18]. Obesity, often present in GDM, also increases the need for insulin therapy during pregnancy and increases women’s vulnerability to T2DM after childbirth [5,19]. The need for insulin increases with a greater metabolic dysfunction—GDM patients receiving insulin have a higher fasting glucose on the oral glucose tolerance test (OGTT) than patients on diet therapy alone [19]. Maternal insulin resistance and increased body mass index (BMI) during pregnancy are associated with changes in placental levels of multiple major metabolites such as lipids, carbohydrates, acylcarnitines, branched-chain amino acids, folate, and choline which correlate, in part, to fetal size [20,21]. GDM, along with its related obesity, contributes to placental DNA methylation, which can negatively affect the developing fetus [1,2,20]. GDM increases the rates of miscarriage, dystocia, cesarean section, neonatal death, premature birth, congenital anomalies, macrosomia, respiratory distress, neonatal jaundice, hypoglycemia, hypocalcemia, and polycythemia [18,22,23]. GDM-related fetal hyperglycemia and hyperinsulinemia are associated with aberrant DNA methylation at the insulin-like growth factor (IGF) 2 and neighboring *H19DMRs*, which increase the risk of developing childhood obesity, neoplasms, and diabetes, in addition to metabolic and cardiovascular diseases during adulthood in children of GDM mothers [1,5,23,24]. More GDM complications occur in women with high pre-pregnancy BMI, currently overweight/obese, and over the age of 35 years [19,22]. Moreover, ethnicity plays a role in metabolic activity [8,9,19]; Asian GDM women need insulin despite their lower pre-pregnancy BMI compared with Caucasian GDM women [19].

Pre-menstrual syndrome [25], prenatal stress [26], post-partum depression [27], and late-life depression/anxiety in menopausal women [28] are examples of mood dysregulations (MDs) that affect women during different stages of life [25]. Studies evaluating MDs during pregnancy report that the prevalence of maternal depression and anxiety is up to 27% and 24%, respectively [29,30] with a high overlap between symptoms of depression, anxiety, and stress [31]. Although mood disorders occur in 7.4% of women during the first trimester, the prevalence dramatically rises as the pregnancy progresses; they reach their highest rates during one year after delivery [32]. Curiously, symptoms of anxiety and depression that develop during pregnancy compared with a pre-pregnancy diagnosis increase the likelihood of not initiating breastfeeding, early termination of breastfeeding, and increased formula usage [33,34]. In addition, prenatal depression correlates with poor maternofetal attachment and a higher postpartum parenting stress [33]. Prenatal and postnatal maternal MDs and perception of stress are associated with prematurity, birth weight abnormalities, stunted growth, cognitive and neurodevelopmental deficits, infants’ temperament, anxiety, and depression, and delay in fine motor and language development [26,27]. Pharmacological treatment of mood disorders during pregnancy mostly comprises selective serotonin reuptake inhibitors (SSRIs) or serotonin-norepinephrine reuptake inhibitors (SNRIs) [32]. However, most psychotropic drugs cross the placental barrier to affect the serotonergic and catechol-aminergic systems in the brain of the growing fetus resulting in long-term neurodevelopmental impairments [32,35]. Longitudinal data show that SNRIs increase the rate of pregnancy hypertension, higher birth weight, and lower Apgar scores in antenatally exposed newborns regardless of the dose or drug class [33]. In addition, sudden cessation of these drugs can result in acute physical and emotional adverse effects [32].

Both GDM and mood disorders can be either initiated or mitigated by food consumption [36,37,38,39]; a hypercaloric diet is reported to alter the endocrine system, promote adiposity, and create an inflammatory milieu that promotes insulin resistance and alters the structure of the brain [36,40]. Experimentally, a hypercaloric diet reduces hippocampal brain-derived neurotrophic factor (BDNF) resulting in the development of depression–anxiety like behaviors in young rats [36]. Frequent consumption of certain food groups, e.g., meat/processed meats, protein of animal sources, and potatoes increased the risk of GDM in women [41]. Similarly, excessive intake of ultra-processed food is associated with increased odds of depression in young adults [42]. On the contrary, moderate to strong pre-pregnancy adherence to a Mediterranean diet, diets that control hypertension, and the Alternate Healthy Eating Index diet reduces the risk of developing GDM by up to 38% [41,43]. In addition, adherence to a Mediterranean diet also reduces depression and mitigates depressive symptoms [44]. Therefore, the literature emphasizes the pivotal role of dietary interventions in GDM and mood disorders [43,44].

Physical activity affects both metabolism and mood [41,45]. Exercise operates through different mechanisms that involve modulation of: (1) gut microbiome alterations and gut dysbiosis [46,47], (2) key signaling involved in vital cellular processes (e.g., autophagy, metabolism, etc.), and (3) signal transduction that counteracts the toxic effects of free radicals and inflammatory mediators [48,49]. Cumulative evidence shows that the odds of GDM decrease by 46%, 30%, and 21% in response to pre-pregnancy exercise >90 min/week, any pre-pregnancy exercise, and early pregnancy exercise, respectively [41]. Sedentary behaviors such as watching television or using computers/internet for a long time increase depression [50]. In parallel, regular exercise lowers the odds of depression, anxiety, and aggression [47,48]. Thus, physical activity represents a key modality for metabolic regulation, especially during pregnancy [38]. Not surprisingly, trivial levels of exercise exert a protective effect against prenatal depression [45].

The co-occurrence of GDM and MDs contributes to the complications alluded to earlier in mothers and their offspring. This review elaborates on the role of diet in causing and improving GDM and MDs. It will also shed light on the benefits of modifying diet, exercise, and intermittent fasting (IF) in a patient suffering GDM associated with MDs.

## 2. Pathophysiology of GDM

Pregnancy is a period of hormonal stress, during which metabolic changes occur to fulfill maternal and fetal needs for nutrients [21,51]. Glucose is the predominant energy source for fetal growth [6]. In normal pregnancy, the utilization of maternal glucose decreases due to a progressive decline in insulin sensitivity (by up to 60% compared with pre-pregnancy levels) lasting until the second trimester [7]. The aim of maternal insulin resistance is to limit maternal use of glucose to ensure its adequate supply to the growing fetus [6]. In this respect, placental lactogen (PL), which acts locally to promote placental growth and function, modulates maternal IGF to ensure adequate fetal nutritional supply [51].

Experimental evidence shows that under normal physiological conditions, PL maintains maternal metabolic homeostasis by promoting the proliferation of pancreatic β-cells [52]. PL also inhibits the production of proapoptotic molecules in the endoplasmic reticulum (ER) stress pathway that are evoked by chronic ER stress resulting in increased human β-cell survival and decreased incidence of diabetes in rodents [53]. However, alterations in PL develop in obesity where vital metabolic molecules become dysregulated, such as adiponectin, an adipose tissue-derived adipokine that contributes to insulin sensitivity and regulates glucose metabolism [54]. Because of hypoadiponectinemia, which is often associated with chronic low-grade inflammation [54,55], PL fails to promote β-cell proliferation [52]. Instead, it exerts a lipolytic activity—increasing circulating fatty acids—resulting in a maternal metabolism shift toward lipids rather than glucose as a source of energy [7]. There is evidence that pregnancy is characterized by mild long-term inflammation with an increase in the neutrophil-to-lymphocyte ratio. Increased maternal age and high pre-pregnancy BMI are associated with exacerbation of inflammation and secondary tissue damage [56]. Pancreatic β-cell exposure to proinflammatory cytokines is associated with ER stress and cellular damage/death [7,53]. GDM develops when pancreatic β-cell insulin secretion is not sufficient to counteract insulin resistance [7,51].

Gut microbiome imbalance has been recently well-acknowledged as a key contributor to serious disorders including diabetes mellitus and obesity [48,55,57], cancer, and a trail of neurodegenerative diseases such as major depression [39,46,58], Alzheimer disease [54], and Parkinson disease [59]. Many factors interfere with the structure of the gut microbiome either by limiting the diversity of intestinal microflora or by promoting the propagation of endotoxic bacterial species. Such factors include unhealthy diet, especially low-fiber/high-fat and high-carbohydrate diets, in addition to physical inactivity and hormonal changes [21,28,60,61]. Microbes detrimental to health act by: (1) replacing health-promoting bacteria e.g., *Lactobacillus*; (2) degrading important nutrients such as amino acids; and (3) producing toxic metabolites that access the systemic circulation to affect the functioning of multiple organs [46,61,62]. Research shows that the maternal gut microbiome has a tendency for a shift to colonization patterns of obesity in normal pregnancy resulting in a heightened inflammatory condition, which is associated with increased insulin resistance [21]. Compared with normoglycemic controls, the gut microbiome of GDM patients exhibits reductions in *Methanobrevibacter smithii*, *Alistipes* species, *Bifidobacterium* species, *Eubacterium* species *Akkermansia*, *Bacteroides*, *Parabacteroides*, *Roseburia*, and *Dialister*, in addition to increased abundance of *Parabacteroides distasonis*, *Klebsiella variicola*, *Ruminococcus*, *Eubacterium*, *Prevotella*, *Collinsella*, *Rothia*, *Desulfovibrio*, *Actinobacteria*, and *Firmicutes*. Changes involving some of these microbes are evident in T2DM patients [63].

Numerous environmental factors interfere with the gut microbiome and maternal metabolome causing pregnancy-related metabolism to shift toward developing insulin resistance and glucose dysregulation [2,21]. For example, consumption of unhealthy food and physical inactivity are lifestyle factors that can alter metabolic homeostasis to a great extent [17,55]. In this regard, excessive carbohydrate intake and exposure to insulin (such as in insulin resistance) trigger acute epigenetic changes in tissues involved in glucose uptake (e.g., skeletal muscle). Such alterations involve decreased DNA methylation in the Ca2+ pump *ATP2A3* gene and accelerated DNA methylation in the gene body of death-associated protein kinase 3 (*DAPK3*) gene, which regulates cell proliferation, apoptosis, and autophagy [40,47]. Studies associate GDM with maternal exposure to endocrine disruptors, environmental chemicals that alter the endocrine system and interfere with hormonal action at different stages resulting in adverse effects [23]. The obesogenic phthalates (diesters of phthalic acid) and bisphenol A (a compound that results from condensation of phenol and acetone) are widely used in the environment and industry. For example, they are included in insecticides and food packaging plastic. These compounds accidently enter the food chain and alter human health. Research links maternal exposure to these compounds to the development of GDM [23].

Pre-pregnancy obesity and excess gestational weight gain increase the risk of GDM [2,21]. Obesity promotes metabolic dysfunction by altering the gut microbiome structure. Meanwhile, dietary lipids interact with molecules involved in the regulation of the immune response resulting in systemic inflammation and secondary insensitivity to insulin [47,49]. Molecules secreted by the adipose tissue and extracellular vesicles, such as adipokines and adipocytokines, interfere with placental nutrient transport [18]. Dysfunctional adipokine profiles associated with excessive accumulation of subcutaneous and visceral adipose tissues are evident in pregnant women with GDM independent of BMI. Adiponectin deficiency promotes insulin resistance, maternofetal hyperglycemia, and DNA methylation [64]. High fat mass is associated with chronic inflammation and persistent production of pro-inflammatory cytokines out of constant activation of macrophages and related toll-like receptor 4 (TLR4) [48,55,61,65]. Constantly elevated levels of cytokines induce various cellular alterations through the modulation of protein properties and functions. These process subsequently are associated with modulation of signal transduction conducive to oxidative damage, increased stress of the ER, and stimulation of mitochondria-related cell death signaling cascades [47,66]. In this respect, cumulative evidence shows that an inflammatory milieu during pregnancy endangers pancreatic islets through the promotion of mutations in pancreatic β-cell factors, paired domain homeobox (*Pax*) family of transcription factors (*PAX4* and *PAX8*), and the chromatin remodeler factor high mobility group protein 20A (*HMG20A*) [1]. Indeed, experimental evidence confirms the contribution of obesity to autophagic failure and heightened apoptosis in pancreatic islets [67].

In addition to the inflammatory effects of maternal obesity, fetal membranes represent an internal maternal environmental factor, which may contribute to GDM and complicate pregnancy through the release of various molecules at different stages of gestation. Although fetal membranes (FMs) of normal glucose tolerance women release detectable levels of high mobility group box 1 (HMGB1), FMs of GDM women produce high level of HMGB1 as depicted by Western blot analysis [5]. HMGB1 is a proinflammatory molecule that increases in T2DM [68] and in GDM during the third trimester [5]. HMGB1 activates signaling involved in inflammation and oxidative stress by binding the receptor of advanced glycation end products (RAGE), and TLR2 and TLR4, which indirectly activate C-X-C chemokine receptor type 4 (CXCR4) [68,69,70]. HMGB1 is a key factor in complications that develop in T2DM such as neuroinflammation and neuropathy [70]. Compared with premature infants with low cord blood HMGB1, neuroinflammation and brain injury is significantly elevated among premature infants with cord blood high in HMGB1 and low in soluble RAGE (sRAGE). sRAGE is a truncated form of the receptor that comprises the extracellular domain of RAGE, and can block RAGE-activation [69].

Women are more genetically prone to metabolic dysfunction than men, and such vulnerability increases during pregnancy [71]. In vitro investigations of epigenetic profiles of human tissues involved in glucose metabolism show higher DNA methylation in genes that regulate energy metabolism, inflammation, and oxidative phosphorylation in women compared with age and gender cross-matched men. These differences were spotted on autosomal chromosomes and the X-chromosome, despite the absence of sex hormones and other factors that differ between sexes [47,71,72,73]. Research has uncovered a genetic overlap of maternal insulin resistance with T2DM involving loci associated with β-cell function [1]. Placental-related activity in GDM is associated with dysregulated expression of micro, long, and circular non-coding RNAs (ncRNAs). Dysfunctional ncRNAs alter cellular metabolic processes as they interfere with β-cell function and body response to insulin. ncRNAs dysregulation is evident in GDM, and it alters the intrauterine environment [17,74]. For instance, long ncRNA *MALAT1* is activated in GDM, resulting in upregulation of the tumor growth factor β (TGF-β)/nuclear factor-kappa B (NF-κB) signaling pathway to subsequently increase the release of inflammatory cytokines [74]. As a result, altered ncRNAs trigger short- and long-term complications in GDM, such as macrosomia, trophoblast dysfunction, and metabolic and cardiovascular diseases. The latter result from the interference of ncRNAs with the expression of target genes in the offspring [17].

## 3. Mood Dysregulations in GDM

Research shows that shifts in feminine hormones increase liability to MDs [28,35]. Evidence signifies that women with GDM are at high risk for developing depression during pregnancy (adjusted odds ratio (AOR) = 2.46, 95% CI: 1.01 to 6.03) [3], and around one-third of women with GDM develop post-partum depression [35,38]. The relationship between mood disorders and GDM is reciprocal [3,35,38]. A prospective Chinese study followed 1426 single-child healthy pregnant women from 8 to 14 weeks of gestation for the development of GDM. Women with high anxiety scores were more likely to develop GDM in analysis adjusted for age, pre-pregnancy BMI, family history of diabetes, gravidity, parity, energy intake, conception, education, occupation, smoking, and alcohol consumption. This relationship was evident in nulliparous women and women under the age of 30 [75]. In fact, observational and genetic investigations show that depression increases susceptibility for obesity and metabolic disorders by promoting mutations in fat mass and obesity associated (*FTO*) and melanocortin 4 receptor (*MC4R*) genes, resulting in alterations in appetite and the hypothalamic-pituitary adrenal axis [76]. Of interest, depressed pregnant women who carry to term give birth to large-size babies compared with undepressed pregnant women [29]. This common feature of GDM and prenatal depression signify a role of maternal metabolism dysregulation in both conditions, which is likely to interfere with the intrauterine environment to affect fetal growth.

The co-occurrence of depression and diabetes is high, and such co-morbidity is closely linked to poor glycemic control, poor self-care ability, functional disability, low quality of life, and premature death. Treatment of depression in diabetics is challenging given the complex nature of diabetes, especially when it occurs during pregnancy [77]. Moreover, reports from observational studies show that GDM women who receive social support and demonstrate self-efficacy are more likely to adopt a healthy lifestyle (e.g., diet and physical activity) following GDM diagnosis. However, integrative knowledge shows that psychological interventions are not integrated in the management of GDM [38]. Conversely, the development of mood disorders during pregnancy may result in a range of adverse effects for mothers such as social maladjustment, marital dissatisfaction, and suicidality [31,35]. In addition, the occurrence of depression in women with GDM is a top predictor of fetal and neonatal adverse effects such as neonatal respiratory distress at delivery (AOR = 3.87, 95% CI 1.32 to 11.35) [78].

Hyperglycemic milieu furthers inflammation and oxidative stress by accelerating non-enzymatic binding of sugars to cellular proteins and lipids, resulting in increased production of highly toxic substances known as advanced glycation end products (AGEs) [46,47,49,55,79]. AGEs activate TLR4 in macrophages and upregulate major signaling pathways involved in inflammatory and oxidative stress by binding their main receptor, RAGE, in addition to numerous other receptors on the cell surface [46,47,66]. Both depression and diabetes are inflammatory disorders. Research confirms that intense activation of nod-like receptor family pyrin domain containing 3 (NLRP3) and its upstream P2X7 receptors is associated with excessive release of inflammatory mediators (e.g., interleukin (IL)-1β and IL-6) [54,77]. The brain is one of the body organs that are affected by inflammatory reactions that follow feminine hormonal changes and hyperglycemia that occur during pregnancy. Such high release of cytokines may damage both neuronal and non-neuronal cells of the brain, which may promote the development of MDs and cognitive changes [28,54,77].

## 4. Dietary Interventions for Gestational Diabetes

Lifestyle interventions of diet therapy and physical exercise, or a combination of both, are the first-line treatment options for GDM [4,51]. These interventions represent a cost-effective way to achieve glycemic control while avoiding insulin therapy [4]. Key benefits of these interventions include decreasing BMI, improving metabolism, reducing the risk of preterm deliveries and low birth weight, and lowering the chance for developing postpartum diabetes [38,80].

Dietary supplements such as polyunsaturated fatty acids (PUFA) [81] and vitamin D [51] have been used to control GDM. A meta-analysis reports that PUFA facilitate glycemic control in pregnant women with GDM [81]. However, PUFA have no effect on macrosomia, neonatal hyperbilirubinemia, preterm delivery, and preeclampsia [81]. Vitamin D is reported to decrease maternal BMI, but its effect on glucose metabolism is trivial [51]. Many reviews report positive effects of specific dietary modifications that target GDM [37,38,80]. In 18 randomized control trials (RCTs) involving 1151 GDM women, modified dietary interventions significantly reduced fasting and postprandial glucose (−4.07 mg/dL, 95% CI −7.58 to −0.57, *p* = 0.02 and −7.78 mg/dL, 95% CI −12.27 to −3.29, *p* = 0.0007, respectively) and reduced the need for insulin therapy (RR = 0.65, 95% CI 0.47 to 0.88, *p* = 0.006). In 16 of these studies, dietary modifications decreased heavy baby birth weight (−170.62 g, 95% CI −333.64 to −7.60, *p* = 0.04) and macrosomia (RR = 0.49, 95% CI 0.27 to 0.88, *p* = 0.02) [80].

Certain dietary patterns are adopted for controlling hyperglycemia of GDM. In low-carbohydrate diets, less than 35–45% of total daily energy intake comes from carbohydrates [82]. Low glycemic index (GI) diet involves the intake of food elements that slowly release glucose; thus, they do not increase blood glucose level. Examples of low GI food include fiber-rich foods and complex unrefined types of carbohydrates (e.g., whole grains and grain bran) [37,63]. Ketogenic diets involve low intake of carbohydrates (<20–50 g or <10% calories from carbohydrates per day) while using fat and protein to complete the needed dietary requirements [82]. Ketogenic diets are used to evoke the production of ketones, acetoacetate, and β-hydroxybutyrate as major alternative energy substrates to glucose [54]. Although the safety of ketogenic diets during pregnancy is not established [82], experimental evidence shows that gestational exposure to ketogenic diets reduces depression and anxiety, and increases sociability in mice offspring [83]. A low GI diet is considered to be safe and the most effective for glycemic control [7,37,63]. In nine RCTs comprising 884 GDM women (mean age = 31.5 years, age range = 28.7–33.2 years, weeks of gestation range = 24.1–30.3), only a low GI diet reduced insulin use (RR = 0.767, 95% CI 0.597 to 0.986, *p* = 0.039) and the newborn birth weight (weight mean differences −161.9 g, 95% CI −246.4 to −77.4, *p* = 0.000) compared with control diets [37]. Furthermore, total energy restriction and low carbohydrate diets had no effect on maternal or newborn outcomes [37,84].

Intermittent fasting (IF) is a religious or cultural practice popular throughout the world. Research has recently proven several health benefits of fasting, and its use as a therapeutic modality [85]. IF in humans is a type of dietary restriction—a term that describes voluntary reduction of nutritional intake except for fluids and basic nutrients [86]. Research shows that dietary restriction is a therapeutic measure that can correct multiple pathologies, promote wellness, and extend lifespan [86,87]. One of the major signaling pathways involved in dietary restriction is IGF-1, which primarily regulates glucose metabolism [88]. Fasting mimetics, such as rapamycin and metformin, are used to influence signaling involved in dietary restriction without actual calorie restriction [87]. Metformin is reported to mitigate symptoms of depression and treat common endocrine disorders among women of childbearing age (e.g., polycystic ovary syndrome) [89].

Caloric restriction, the most common form of dietary restriction, involves reducing the daily intake of calories by 20–30% with an ad libitum intake of water [87]. The difference between fasting and caloric restriction is that fasting entails completely refraining from food for extended periods (up to 18 h per day), and may take place over the course of several days or weeks [67,90]. There is less emphasis on the caloric content of food ingested during non-fasting hours [67]. Acute fasting accelerates the level of acyl-ghrelin, known as the orexigenic gastrointestinal hormone, which functions as a ligand for the growth hormone secretagogue receptor in the brain. This effect contributes to some of the neuroprotective activities of fasting, particularly increasing the level of neurogenic transcription factor Egr-1 in the dentate gyrus resulting in hippocampal neurogenesis [85].

Findings on caloric restriction during pregnancy are mixed. Although reduction of carbohydrate intake may induce ketonemia, a recent RCT shows that a slight reduction in maternal carbohydrate intake (135 g/d) does not result in detectable differences in blood ketones compared with usual carbohydrate intake [84]. A low carbohydrate diet had no effect on maternal and fetal GDM-related adverse effects. Moreover, nonadherence to this diet was reported in 65% of the participants [84]. Experimental evidence shows that 20% caloric restriction in pregnant rats in addition to micronutrient supplementation (for the control of malnutrition) does not decrease litter size and birth weight or delay development. Offspring of caloric restricted rats exhibited earlier sucking reflex, earlier response to the negative geotaxis, and better locomotor activity. In adulthood, these offspring were leaner and had less preference for palatable food, indicating less obesity tendency. No alterations in memory were depicted [91]. In another animal experiment, 50% prenatal caloric restriction reduced maternal plasma choline, betaine, and S-adenosylmethionine (SAM) by 40%, 45%, and 20% compared with the control diet. These compounds originate from one-carbon metabolism, which occurs in the liver [20]. Folate and choline are key nutrients that regulate fetal cell division and fetal development. SAM is used by various methyltransferases as a methyl donor to generate S-adenosylhomocysteine (SAH), which converts into homocysteine, which is involved in methylation reactions [20]. However, SAH and homocysteine were not affected by caloric restriction. These findings show that in the absence of micronutrient supplementation, exaggerated gestational caloric restriction limits fetal nutrient supply with possible implications for reducing birth weight and increasing the risk of cognitive dysfunction and spina bifida. Interestingly, all of these changes were not detected latterly in first generation pups during pregnancy, indicating that minor epigenetic changes induced by caloric restriction do not trigger persistent metabolic changes in the progeny [20]. It is noteworthy that a reduction of total caloric intake (799 kcal/day) by Tanzanian Maasai women during the third trimester is associated with the occurrence of low birth weight (<2500 g) in one third of the neonates [92]. Therefore, methods that involve regulation of food intake while maintaining adequate caloric intake would be safer than excessive caloric restriction. However, rigorous evidence is required to confirm the effectiveness of such methods.

## 5. Case Report

A 36-year-old woman with a pre-pregnancy BMI of 21.7 and a family history of T2DM (father and brother) was diagnosed with GDM. This was her third pregnancy, and she had no history of GDM in earlier pregnancies. As her pregnancy progressed, she began to develop a substantial craving for carbohydrate-rich foods, such as bread, pasta, and rice. Her body weight, abdominal circumference, and height of the uterus increased steadily (Figure 1). She complained of severe back pain directly underneath the scapula and leg cramps. The former was persistent while the latter were most disruptive during sleep. She had difficulty in sleeping because of frequent urination and inability to return to sleep. Her sleep quality was poor, and she complained of prolonged dreams. When she got up, she felt exhausted and depressed. She experienced headaches in the morning that were not associated with changes in blood pressure. Her routine urine consistently tested negative for glucose, ketone, and albumin. Since the first week of pregnancy, the size of the fetus was slightly large for its gestational age. The doctor suggested that such an increase was an indicator of good intrauterine growth or due to improper calculation of the last menstrual period. During the 26th gestational week, the physician noted that the weight, in addition to the circumferences of the head, thoracic cavity, and abdominal cavity of the fetus, were noticeably large (Table 1 and Figure 2a), suggesting a possibility of GDM. The patient immediately refrained from the intake of refined carbohydrates although she continued to consume three main meals and light snacks. Although her random plasma glucose was not elevated (88 mg/dL), her serum glucose levels were high in an OGTT: fasting, 30 min, 1 h, and 2 h glycemia levels were 86, 210, 189, and 163 mg/dL, respectively. After consulting a GDM specialist, she was given a glucometer for glucose self-monitoring before and 2 h after meals. Her self-management plan was refined during the 28th gestational week to include further dietary modifications (consumption of green salad, bee honey, jelly beans, check peas, yoghurt, vegetable soup, fish, eggs, soy, tofu, and fermented soy) and walking for 15–20 min after each meal, in addition to refraining from refined carbohydrates. Subsequently, her leg cramps disappeared and her 2 h post prandial glucose level was under 120 mg/dL, which was intended as a target for glycemic control (Figure 3). However, exercise and a low-caloric/low GI diet were not always sufficient to maintain her 2 h post prandial glucose level under 120 mg/dL, particularly after lunch and dinner (Figure 3), and fetal growth remained abnormal (Table 1). Failure to obtain these results heightened the patient’s tension and distress, which were associated with greater loss of control over her blood glucose. For example, in some instances she ate low-calorie foods, but her 2 h post-prandial glucose was above 120 mg/dL, and after walking, her glucose level rose further. The patient worried about the fate of herself and her pregnancy (patient’s narratives regarding her psychological experience of GDM are described in the Appendix A). Accordingly, she decided to maintain a long interval of fasting between meals. She consulted the diabetes clinic, and the doctor confirmed that she should meet her daily dietary requirements (2200 kcal/d) regardless of the timing (Figure 4). Accordingly, she consumed two main meals per day, 13–15 h apart. Fasting started around 2:00–4:00 a.m. after an early breakfast. Then, she refrained completely from eating and drinking until around 5:00 p.m.—the time of her dinner. During the non-fasting period (5:00 p.m.–4:00 a.m.), she drank a large quantity of plain fluid and ate light snacks. IF, in addition to exercise and caloric restriction, maintained her 2 h post prandial glucose level around 120 mg/dL. However, the anti-hyperglycemic effects of IF, dietary modification, and exercise remarkably diminished during the last week of pregnancy (Figure 3d,e), which was associated with slight deviations in fetal growth parameters (Table 1).

As shown in Table 1 and Figure 2, maternal hyperglycemia resulted in fetal measures that were 1.2 (femur length) to 3.1 (bi-parietal diameter) standard deviations above the average (panel a). However, physical exercise, dietary modifications, and IF adopted by the patient resulted in a remarkable control of fetal growth (Table 1, Figure 3, panel b and c). Because the patient had a history of delivery by Cesarean surgery (C-section), this baby was also delivered by C-section under the effect of spinal anesthesia. There was no abnormal bleeding and the patient did not receive a blood transfusion. This pregnancy resulted in a 48 cm-long baby weighing 3150 g, with head and chest circumferences of 35.5 and 33.5 cm, respectively.

Table 2 shows that after 7 weeks of dietary modifications (which include 3 weeks of IF), patient’s fasting blood glucose level, liver and kidney profiles, and electrolytes were all within normal ranges. Slight reductions in red blood cell count, hemoglobin, and hematocrit indicate the presence of mild anemia, which commonly occurs during pregnancy. Plasma concentrations of total proteins were slightly reduced, whereas levels of C-reactive protein and creatine kinase were extremely high (discussed in detail below).

Although patient’s sleep quality did not improve after exercise/dietary interventions/IF, the frequency of urination at night decreased, which was associated with prolonged sleep time and less frequency of experiencing depression. The patient reported that decreasing the number of main meals from three to two during IF was a significant relief: she felt that she was relieved from the effort of controlling her blood glucose after the lunch meal because this meal was omitted, and her 2 h post-prandial glucose level after dinner could be better maintained within the pre-set target (less than 120 mg/mL). The shift in fetal growth toward normal levels considerably alleviated worries about the health of her newborn.

## 6. Discussion

This report shows that although urine consistently tested negative for glucose/ketone bodies, and fasting/random blood sugar were not elevated, insulin resistance was present, which negatively affected the intrauterine environment as portrayed by increased fetal body size. Following an exercise and dietary regimen for sensitizing the body to insulin resulted in several positive effects. Figure 2 shows that the 2 h post prandial blood glucose level was maintained within the reference range, with the exception of a small number of instances. As shown in Figure 1, maternal weight increased from 64.9 to 72.1 kg between the 13th and 32nd weeks. However, IF, dietary modifications, and physical activity resulted in considerable improvements: during the period between the 32nd and 36th weeks, her body weight dropped from 72.1 to 68.3 kg, and the abdominal circumference decreased from 95 to 92 cm between the 32nd and 34th weeks. The decrease in the height of the fundus corresponded to the obvious change in fetal weight. It is important to note that a decline in maternal weight and abdominal circumference took place only after starting IF.

High intake of carbohydrates is reported to induce insulin resistance by evoking epigenetic changes in skeletal muscle (one of the major body tissues involved in glucose uptake) involving increased DNA methylation in the gene body of *DAPK3*, which modulates processes involved in cellular functioning and survival [40]. In this patient, increased craving for carbohydrate-rich foods was associated with an increased body weight, abdominal waist circumference, and plasma glucose, in addition to sleep disturbance and psychological distress. By comparison, IF, reducing the consumption of carbohydrates, and adopting a diet rich in vegetables and proteins, were associated with remarkable reductions in blood glucose, body weight, waist circumference, height of the fundus, and measures related to fetal growth. The patient felt less distressed about controlling her physiological alterations.

Dietary fibers also play a role in the activation of signaling pathways involved in dietary restriction, such as IGF-1, which plays a key role in the regulation of glucose metabolism [88]. Vegetables, especially leafy vegetables, are rich in dietary fibers, which are known to promote the growth of healthy bacteria in the gut [39]. In addition, the surface of fresh vegetables contains probiotic bacteria such as lactic acid bacteria—around 35% of these bacteria can survive gastric conditions and reach the gut to exert therapeutic effects [94]. Furthermore, research signifies the contribution of the complex carbohydrate content of low GI food to glycemic control via a complex mechanism that involves the regulation of gut microbiota [47,63]. Such foods provide health-promoting intestinal bacteria with the appropriate type of dietary fibers. As a result of good microbial fermentation, the levels of short-chain fatty acids (e.g., acetate, propionate, and butyrate) increase [63]. These compounds activate the production of gut hormones (glucagon-like peptide 1 and peptide YY) and leptin, leading to increased satiety and improved response to circulating insulin [47,63].

This patient increased her intake of protein, such as soy, check peas, fish, egg, and yoghurt. The literature indicates that protein intake modulates insulin activity and glucose metabolism in men and nonpregnant women. Although findings on the association between protein consumption with insulin sensitivity during pregnancy are equivocal, the quality of dietary protein (e.g., of plant origin) is associated with improved insulin sensitivity [2,46]. A high-protein diet results in increased supply of fermented proteins by colonic microbiome leading to increased production of protein fermentation end-products, such as polyamines [95]. Experimentally, the intake of soy and fermented milk proteins, especially in the morning, positively alters the structure of gut microbiota and results in increased production of health promoting nutrients (e.g., short-chain fatty acids) by bacteria of the colon [96]. Several lines of evidence show that soy foods are low GI, and they can lower blood glucose level in healthy adults [97,98] and in T2DM patients [99]. The anti-hyperglycemic effects of soy appear to be gender specific. In this regard, a longitudinal Japanese study involving follow up of 13,521 participants over 10 years documents protective effects of high soy consumption against T2DM only among women [97].

Muscular mitochondrial dysfunction may play a major role in the development of insulin resistance [40,46,47,49,100]. However, evidence also indicates that the intake of foods rich in soluble milk proteins increases the fractional synthesis rate of mitochondrial muscle protein regardless of the amount of consumed protein [101]. A recent investigation revealed that adherence to the Mediterranean diet for one year increased the abundance of specific gut bacterial species that are associated with decreased inflammation (detected by low levels of C-reactive protein and IL-17) and frailty in European elderly [102]. The high intake of protein-rich food expressed in the current case suggests a supporting role of protein intake in insulin sensitivity during pregnancy. Fermented dairy products such as yoghurt are also a rich source of lactic acid bacteria, which are commonly used as probiotics to correct systemic inflammation through modulation of the gut microbiome structure [46,59,103]. Furthermore, evidence shows that human consumption of yoghurt beverage fortified with lactotripeptides increases circulating levels of W-containing peptides [103]. W is an essential amino acid known as tryptophan. Enzymatic activities (proteolytic/peptidolytic) that affect milk proteins generate W residues either as a free amino acid or as part of peptide sequences. W promotes protein synthesis and acts as a precursor of major biomolecules, such as serotonin, melatonin, tryptamine, niacin, nicotinamide adenine dinucleotide (NAD), phosphorylated NAD (NADP), quinolinic acid, and kynureric acid [103]. These molecules play key roles in the regulation of oxidative stress, inflammation, and metabolism, which are necessary for proper physiological functioning [47,59,87]. W-containing peptides inhibit angiotensin converting enzyme (ACE) [103], which is involved in human acquisition of the current, highly-infectious corona virus disease 2019 [47,48,49]. W-containing peptides also demonstrate strong antioxidant, antidiabetic, neuroprotective, and satiating-related activities [103].

Notably, levels of C-reactive protein (a marker of inflammation) and creatine kinase (a marker of muscle dystrophy) [46,49,54,55] were extremely high in the case-study patient (Table 2), indicating that inflammatory reactions associated with GDM persist even when blood glucose level is controlled. The patient’s high levels might explain the leg cramps and back pain experienced before the commencement of diet and exercise. In fact, the level of C-reactive protein is significantly higher in GDM women compared with their healthy counterparts, indicating that C-reactive protein may be a marker for early diagnosis of GDM [104]. Although the patient’s diet contained sufficient amounts of animal- and plant-derived protein, her serum total protein and albumin levels were slightly low. Such a reduction may be justified by anabolic resistance secondary to insulin resistance, albeit not obviously [47,55]. Alternatively, the increased creatine kinase level may have been caused by destruction of proteins (e.g., in skeletal muscle) to increase energy supply [47,105]. Unfortunately, C-reactive protein and creatine kinase were not estimated either before the dietary treatments or before childbirth, which makes the effect of dietary modification and IF on GDM-related inflammatory and catabolic processes unclear. Accordingly, assessing the effect of dietary modifications and IF on these parameters in future studies would advance GDM-relevant knowledge.

Bee honey is a potential source of natural antioxidants, such as flavonoids, phenolic acids, and terpenoids. It demonstrates a capability of counteracting the effects of oxidative stress underlying the pathogenesis of various diseases, such as T2DM, neurodegenerative disorders, cancer, and atherosclerosis [39,106,107]. Bee honey can inhibit the growth of pathogenic bacteria through two mechanisms. It demonstrates an antiseptic activity through its relatively high internal contents of hydrogen peroxide and phenolic acids [106]. It is also rich in some strains of lactic acid bacteria that exhibit a strong probiotic potency, such as *Lactobacillus plantarum*. Tests in simulated gastric juice signify the capacity of these bacteria to survive in conditions with highly acidic pH. Thus, they can withstand gastric and bile secretions and remain viable until they reach the intestine. They express strong antipathogenic properties due to their hydrophobic, auto-aggregative, and co-aggregative abilities. They can promptly foster bacterial adhesion and colonization on the host intestinal tract due to their efficient production of exopolysaccharide [108].

Although the composition of gut microbiota of this patient was not examined before or after the dietary modification, fecal properties were exceptionally different before (foul-smelling and foamy in texture) and after dieting and IF (less foul-smelling and having a soft-creamy texture). Examination of the diet adopted in this case appears to indicate that refraining from the intake of carbohydrate- and fat-rich food, in addition to increased consumption of vegetables and protein-rich food, can positively affect the composition of the gut microbiome. Several lines of evidence show that correction of gut microbiota composition corrects dysbiosis and alters inflammation by inhibiting the passage of bacterial toxins from the gut to the main circulation [46,54]. Figure 5 provides an illustration of how food consumed by the patient may correct GDM and associated MDs.

Although dietary modification, exercise, and IF exerted an anti-hyperglycemic effect in our patient, this effect tended to fade with the progress of pregnancy (Figure 3d,e). The last gestational week witnessed increases in maternal weight, waist circumference, and height of the uterus (Figure 1). The latter was associated with fetal weight gain (Table 1). Insulin resistance in late pregnancy is hormone-driven [51,109]. Available knowledge shows that dietary supplements (e.g., vitamin D) and physical activity during pregnancy may impede maternal weight gain, but fail to control insulin resistance with the progress of pregnancy due to the strong activity of placental hormones, which change significantly in late pregnancy [51]. Alterations in placental hormones contribute to the development of GDM and increase its severity, especially in late stages [51,109]. Several mechanisms have been proposed. Deficiency of placental sex hormone-binding globulin (SHBG) alters insulin signaling by upregulating the activity of extracellular signal-regulated kinase (ERK), resulting in adverse reactions in the placental and fetal environment, such as suppression of proliferation and increased apoptosis [110]. Placental variant growth hormone (GH-V) promotes placental growth. Nonetheless, its over-production modulates maternal IGF-1, resulting in peripheral insulin resistance. Alternatively, expansion of maternal pancreatic β-cell mass is associated with insulin availability, which blocks GH-V [51]. From another perspective, estrogen during pregnancy modulates its receptors to stimulate insulin biosynthesis and secretion by pancreatic β-cell, and to also increase sensitivity of tissues involved in glucose uptake to plasma insulin [109]. However, estrogen dysfunction in GDM is associated with oxidation of estrogen by cytochrome P450 enzymes. Oxidative metabolism of estrogen results in genotoxic metabolites such as 2-hydroxyestrogen and 4-hydroxyestrogen. These metabolites exist in the blood, and induce DNA damage in various tissues. They also bind insulin, neuroglobin, human serum albumin, and immunoglobulin. Their binding to insulin, a process known as insulin estrogenization, hinders insulin affinity of binding to insulin receptors, resulting in vivid insulin resistance [109]. Given the role of pregnancy related hormones in glucose metabolism dysfunction, interventions that trigger insulin secretion may be promising for GDM prevention and control [51].

Experimental research shows that IF alleviates hyperglycemia in obese mice that continue to receive a high-fat diet [67]. The underlying mechanism involves restoring autophagic flux in β islets, promoting pancreatic regeneration as indicated by increased nuclear expression of *NEUROG3*, increasing β-cell survival and pancreatic islet mass, and improving glucose tolerance by boosting glucose-stimulated insulin release [67]. Pancreatic islets and extra-islet endocrine clusters contain multi-lineage potential progenitors. Although these cells contain insulin, they are glucose unresponsive because of their poor expression of glucose transporter 2 (Ins^+^Glut2^LO^ cells). However, metabolic stress triggers the differentiation of Ins^+^Glut2^LO^ cells into mature and functional β-cells [51]. This could possibly be a mechanism for increasing pancreatic islet mass by IF, albeit investigations are needed to confirm the plausibility of this scenario. In addition to the direct effects of IF on insulin production, its effects on body composition, which entail discouraging fat deposition, indirectly alleviate metabolic dysfunction through correction of sensitivity to insulin [67,87]. A current meta-analytic review shows that diurnal IF during Ramadan significantly changes body composition in a positive manner, reducing obesity in individuals aged 16 years and above [111]. Similarly, a recent RCT prompted post-menopausal women to fast from 8 p.m. to 12 a.m. (the next day) for 6 weeks. The findings of this study revealed a considerable reduction in fat mass (an average of around 2 kg) with no adverse effects on muscle mass as indicated by dual-energy X-ray absorptiometry [90]. It is documented that obesity in old age is a key factor for insulin resistance and frailty [46,47]. Obesity in menopausal women is associated with high distress, and mediates the effects of vasomotor symptoms—the cardinal symptoms associated with the drop of estrogen (the main feminine hormone)—on symptoms of depression, anxiety, and cognitive decline in these women [28].

Our patient complained of insufficient night sleep and poor sleep quality, which were associated with headache and dysphoria in the morning. Poor sleep in reproductive-age women results from alterations in the circadian rhythm and melatonin reduction that are associated with hormonal fluctuations over different phases of the menstrual cycle. Poor sleep causes mood symptoms among these women [25]. Moreover, sleep disturbances contribute to circadian dysfunction, metabolic dysregulation, and the development of T2DM [112]. It is possible that the metabolism of this patient was altered as a result of a sleep-disordered breathing problem, which is common in pregnancy [113,114]. In nine studies, pregnant women with sleep-disordered breathing expressed a three-fold increase in the occurrence of GDM after adjusting for BMI (95% CI 1.89–4.96) [113]. Obstructive sleep apnea (OSA) is a complex sleep disorder of fragmented sleep and intermittent hypoxia that results from repetitive episodes of upper airway closures or partial collapse during sleep [112,114]. OSA prevails in 52.4% of obese pregnant women with diet-controlled GDM at a median gestational age of 29 weeks. OSA and sleep parameters related to oxygen desaturation significantly correlate with higher fasting insulin resistance and higher β-cell dysfunction in GDM [114].

Evidence indicates increased nighttime awakening with the development of headaches the next day [115]. Serotonin (5-HT) is a neurotransmitter that plays major roles in diverse human behaviors, including circadian rhythm, mood regulation, sleep, appetite, sexual function, and pain [28,54,115]. Alterations in serotonin may be a possible cause of these complaints. Dysregulation of the subcortical serotoninergic system, particularly in the dorsal raphe nucleus in the pons and midbrain where most production of brain serotonin occurs, is associated with alterations in the transition from non-rapid eye movement sleep (NREM) to REM sleep. This is because initiation of REM sleep (the micro-architecture of sleep) requires cessation of serotonergic neuron firing, and decreased cycling within REM sleep leads to headaches (e.g., migraine) [115].

Although the patient’s concerns about fetal abnormalities persisted, she felt some relief that fetal growth shifted toward normal levels in a relatively short time. In addition to the psychological effect of these events, another possible reason for reduction of her distress was the effect of exercise and healthy foods (protein-rich food and vegetables), both on her gut microbiome and brain [46,54,59]. Evidence denotes that certain dietary elements, such as fermented milk and bee products, can significantly affect the brain in a positive way by downregulating signaling involved in inflammation and oxidative stress [54,59,103]. In particular, yoghurt consumption provides the body with tryptophan, the main precursor of serotonin [103], which is the main neurotransmitter involved in the regulation of mood and sleep [39,107]. Moreover, this patient consumed natural honey as a source of sugars. Bee honey shares many biological properties with other bee products, such as royal jelly and its lipids, which can modulate the activity of estrogenic receptors in the brain [54,116]. These receptors play a major role in the regulation of insulin signaling [109], and in signal transduction (e.g., of serotonin and acetylcholine) involved in cognitive functioning and mood regulation [28]. Moreover, cumulative knowledge indicates that bee honey increases brain levels of BDNF, resulting in improvement of mood and alleviation of depression [107].

Additionally, emotional relief in this case may also be judged as an effect of IF. In fact, several lines of evidence show that dietary restriction and IF exert antidepressant activities [85,87,117]. The underlying mechanisms are complex, and involve increasing phosphorylation of cyclic adenosine mono phosphate-response element-binding protein (CREB), thus increasing brain stress tolerance, correcting cerebral metabolism and improving neuronal supply of ketones as alternatives to glucose, activating neuronal autophagy, increasing the production of BDNF, increasing levels of heat-shock protein, suppressing neuroinflammation, and correcting mitochondrial dysfunction resulting in less generation of free radicals and less neuronal apoptosis [85,87]. Mood regulation as an effect of IF in humans is closely associated with fasting effects on body composition, i.e., weight loss in obese people [117]. Research emphasizes safety of fasting as a therapeutic practice [85,117]. However, to our knowledge, this is the first study to report positive effects of IF on GDM and MDs during pregnancy, and no adverse fetal or maternal effects occurred.

This report signifies the potential maternal and fetal benefits of dietary modifications, exercise, and IF practice in patients with GDM. However, this study has several limitations. It is primarily based on a single case, which is not an ideal one: GDM treatment was chosen and implemented by the patient herself, the patient’s mood was described based on her narratives (not assessed by a specific measure), and not all needed data are available (e.g., fetal growth measures and laboratory investigation before and after dietary interventions). A rigorous evaluation of the effects of IF as a modality to control GDM and associated MDs in a large cohort would be valuable.

## 7. Conclusions

Insulin resistance during pregnancy affects maternal body composition and psychological well-being, in addition to fetal growth, even at slightly elevated levels of blood glucose. Dietary measures including IF, caloric restriction, and exercise, can increase body sensitivity to insulin, reduce body fat deposition, and correct abnormalities of fetal growth, which are associated with hyperglycemia. Positive effects on maternal mood may be attributed to the reduction of inflammatory markers following these interventions. Further investigations are needed to evaluate the cost-effectiveness of IF as a treatment of GDM and mood disorders in pregnancy.

## Figures and Tables

**Figure 1 ijerph-17-09379-f001:**
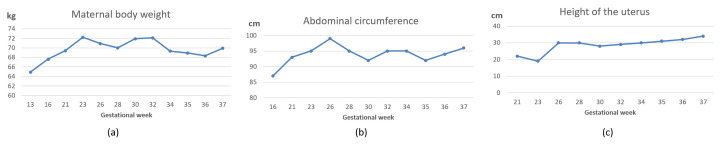
Overall changes in maternal body weight (**a**), abdominal circumference (**b**), and height of the uterus (**c**) before and after dietary modifications/intermittent fasting. Dietary modifications were implemented during the 26th week and intermittent fasting practice started at the 30th week and continued until the end of the pregnancy.

**Figure 2 ijerph-17-09379-f002:**
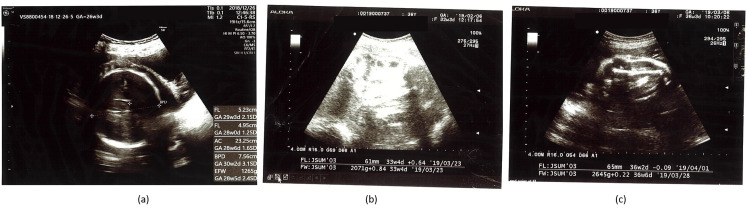
Changes in fetal measures after dietary modifications and intermittent fasting. As shown in panel (**a**) hyperglycemia caused all fetal parameters to be significantly greater (ranging between 28 weeks to 30 weeks) than the real gestational age (26 weeks). Exercise and caloric restriction remarkably ameliorated fetal weight gain resulting in an almost normal fetal weight, panel (**b**). Intermittent fasting in addition to caloric restriction and physical activity resulted in a normal fetal weight, panel (**c**). Most other fetal growth measures tended to be within normal range (Table 1). Pictures were edited to remove patient’s identifying data.

**Figure 3 ijerph-17-09379-f003:**
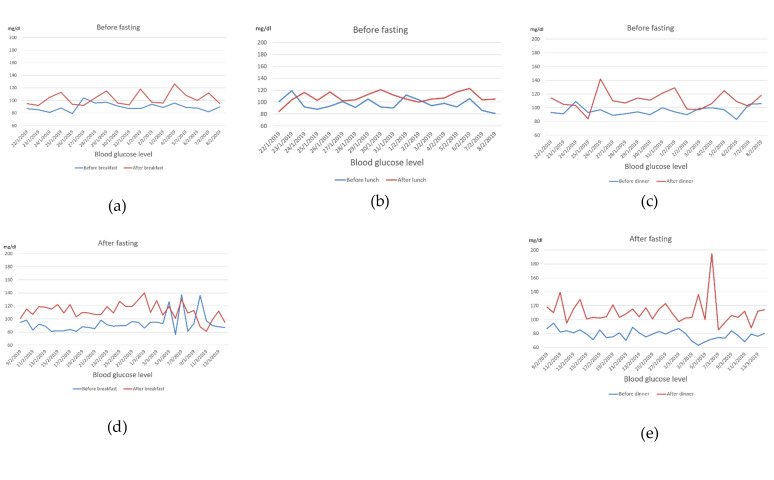
Maternal blood glucose level before and 2 after meals within the context of dietary modifications, exercise (**a**–**c**), and after intermittent fasting (**d**,**e**) in late pregnancy. Dietary modifications were implemented during the 26th week and intermittent fasting practice started at the 30th week and continued until the end of the pregnancy. The lunch meal was dropped after the commencement of intermittent fasting.

**Figure 4 ijerph-17-09379-f004:**
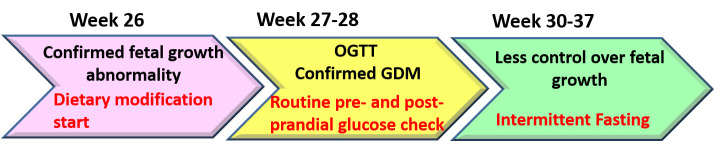
A time-sequence illustration of gestational diabetes mellitus (GDM) diagnosis, change in fetal growth, and the use of dietary modifications/intermittent fasting interventions. During the 26th gestational week, exaggerated fetal growth suggested GDM occurrence. The patient immediately refrained from the intake of refined carbohydrates. An oral glucose tolerance test (OGTT) was performed during the 27th week. The patient practiced light exercise in addition to low carbohydrate consumption. At the end of the 28th week the patient started recording her blood glucose level on a daily basis directly before each meal and 2 h after dining. Fluctuations in the 2 h post-prandial glucose level, in addition to the persistence of abnormal fetal growth, forced the patient to medicate her GDM through intermittent fasting starting from the 30th gestational week. During the fasting course, the patient retained exercise and the previous modification in diet quality.

**Figure 5 ijerph-17-09379-f005:**
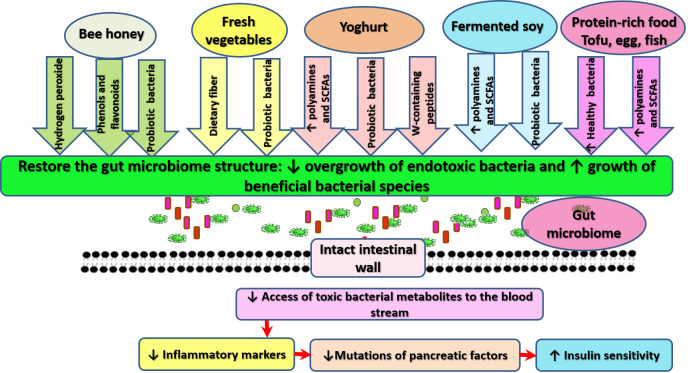
Possible mechanisms through which foods consumed by the patient can promote the gut microbiome and decrease insulin resistance. The dietary pattern adopted in this case may operate through various means to positively alter gut microbiome structure, inhibit intestinal wall leakage, and prevent associated inflammatory response, which may offer protection for pancreatic islets against inflammation.

**Table 1 ijerph-17-09379-t001:** Changes in fetal growth parameters following dietary modification, exercise, and intermittent fasting.

Gestational Age	BPD (mm)	HC (mm)	AC (mm)	FL (mm)	EFBW (g)
26w3d	76 + 3.1 SD 97.5%	--	233 + 1.6 SD 95%	52 + 2.1 SD 97%	1265 + 2.4 SD 97%
27w5d	75 + 1.6 SD 90%	--	242 + 1.3 SD 90%	56 + 2.3 SD 97%	1421 + 2.0 SD 97%
30w1d	82 + 1.83 SD 90%	301 ▲ 97%	275 + 1.9 SD 95%	60 + 1.9 SD 97%	1947 + 2.4 SD 97%
32w3d	87 + 1.8 SD 90%	--	274 + 0.65 SD 50%	61 + 0.64 SD 50%	2071 + 0.84 SD 97%
34w1d	91▲ 50%	339 ▲ 97%	290 + 0.6 SD 10%	67 + 1.5 SD 90%	2480 + 1.2 SD 50%
36w3d	93 + 1.47 SD 50%	--	303 + 0.4 SD 10%	65 − 0.09 SD 10%	2645 + 0.22 SD 50%
37w3d	95 + 1.5 SD 50%	--	302 − 0.02 SD 10%	73 + 1.75 SD 90%	2886 + 0.44 SD 50%

BPD: bi-parietal diameter; HC: head circumference; AC: abdominal circumference; FL: femur length; EFBW: estimated fetal birth weight; W: week; d: day; SD: standard deviation; --Values of the parameter were not recorded in the patient’s record; ▲SD values were not recorded in the patient’s record. N.B. Percentiles are reported according to Papageorghiou et al. [93].

**Table 2 ijerph-17-09379-t002:** Maternal biochemical and hematological parameters during the 34th week of gestation (3 weeks after intermittent fasting and 7 weeks after dietary modifications).

Parameter	Patient’s Value	Reference
Fasting blood sugar	87 mg/dL	70–110
Bilirubin	0.43 mg/dL	0.20–1.20
Aspartate aminotransferase	22 IU/L	8–38
Alanine aminotransferase	14 IU/L	4–44
Total protein	6.3 g/dL	6.7–8.3
Albumin	3.2 g/dL	3.6–5.2
Creatine	0.51 mg/dL	0.40–0.80
Sodium	135 mEq/L	137–147
Potassium	4.2 mEq/L	3.5–5.5
Chloride	106 mEq/L	97–108
White blood cells	83 × 10^2^	40–85
Red blood cells	375 × 10^4^	380–480
Hemoglobin	11.0 g/dL	12.0–15.0
Hematocrit	32.4%	34.0–45.0
Activated partial thromboplastin clotting time	23.4 s	25.0–40.0
C-reactive protein	0.64 mg/dL	0.00–0.30
Creatine kinase	264 IU/L	43–165
Urine ketone bodies	Negative	Negative

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
