# Peer review of "Intermittent Fasting, Dietary Modifications, and Exercise for the Control of Gestational Diabetes and Maternal Mood Dysregulation: A Review and a Case Report"

_ijerph, 2020, doi:10.3390/ijerph17249379_

Round 1
Reviewer 1 Report
This short review and case report is focused on the possible beneficial effects of intermittent fasting on metabolic and affective alterations that occur in pregnancy complicated by GDM. Overall, the topic is interesting, as the first therapeutic approaches in GDM is medical nutritional therapy and novel dietary patterns are needed to achieve good glycemic control and prevent GDM complications. Furthermore, nowadays there is growing interest in fasting practice for the control of metabolic and other diseases. However, I believe that this work might be considered for publication only after an extensive revision. Specifically, the authors should address the following issues:
Major comments
- The first part of this review (introduction section) is not well organized. More specifically, information and evidence about different subjects (i.e. GDM overview, role of gut microbiome in the pathogenesis of GDM and GDM complications, mood dysfunction in GDM) are reported in a single paragraph in a random order. The authors should make an effort to reorganize this section following a more precise scheme. More in details, I suggest creating a paragraph (introduction) including lines 41-58 + lines 135-161 (overview of GDM epidemiology, pathophysiology, complications, therapy etc); then a paragraph on the role of gut microbiome in GDM pathophysiology etc.; a paragraph focusing on mood disorders and GDM etc.; and, lastly, a paragraph focusing on fasting practice.
- In the introduction section, besides the epidemiologic data, the authors should mention also the main complications of GDM (Acta Diabetol. 2018 Dec;55(12):1261-1273. doi: 10.1007/s00592-018-1208-x; J Endocrinol Invest. 2018 Jun;41(6):671-676. doi: 10.1007/s40618-017-0791-y).
- Line 54. The authors should briefly mention the role of epigenetic and environmental factors in the development of GDM and GDM complications (Int J Mol Sci. 2020 Jun 4;21(11):4020. doi: 10.3390/ijms21114020; Nutrients. 2020 Feb 19;12(2):525. doi: 10.3390/nu12020525).
- Lines 114-117. It is worth reporting that during pregnancy not only adipose tissue but also gestational tissues contribute to the increase of pro-inflammatory cytokines and other molecules, which might favour the onset of GDM and the development of GDM complications (Acta Diabetol. 2019 Jun;56(6):681-689. doi: 10.1007/s00592-019-01304-x).
- Lines 149-162. In this section, the authors discuss the main dietary approaches in GDM. Certain dietary patterns (LGI diet, ketogenic diet) are reported as commonly adopted for the control of GDM. However, as yet, there is no consensus on which specific nutritional approach should be used in terms of total energy intake and macronutrient distribution. Moreover, the adoption of a ketogenic dietary approach is, to date, not recommended in pregnancy due to the lack of safety clinical data. The authors should clarify this concept, underlying that, overall, caloric restriction is not recommended during pregnancy, even in GDM, because it may have adverse effects on fetal birth weight. As for LGI diet, the authors should add that a considerable body of research suggests that the LGI diet is reasonably safe and might have positive effects on certain metabolic outcomes in GDM. By contrast, the LGI diet has not shown to have a marked influence on other obstetric maternal or fetal outcomes (body weight gain, birth weight, proportion of LGA, and macrosomia) so far (Nutrients. 2019 Jul 9;11(7):1549. doi: 10.3390/nu11071549).
- In the case report section, relevant clinical data are missing or incomplete. Specifically, it is important to report pre-pregnancy BMI, blood glucose values at OGTT, fetal ultrasound parameters (head circumference, waist circumference, fetal weight) and, more importantly, the percentiles of these parameters.
- Lines 190-192 and 196-199. I believe that it is not necessary to report the personal opinion of the doctor, nor what the doctor told the patient. A case report should give objective data and information.
- Lines 203-205 and lines 213-215. Details about the dietary plans should be given, including calories/day, percentages of macronutrients (carbohydrates, lipids, proteins), number of meals etc. In the same way, the description of the “fasting” dietary plan, which is the main point of this review, is too general and synthetic. I suggest going into details.
- Lines 208-209 and lines 226-227. Please, specify the percentile of fetal weight at this stage. Objective parameters are needed. Percentiles of biometric parameters should be reported to make comparisons and evaluate the effects of this dietary plan.
- Line 207. Pre-prandial and post-prandial glycaemic targets should be indicated. Which targets have been employed to establish the degree of glycaemic control?
- Lines 224-228. There is no information about glycaemic control after intermittent fasting dietary plan: did the patient achieve the prefixed glycaemic targets? What about the biochemical parameters (blood glucose, kidney function, electrolytes, urine ketones etc) during fasting intermittent diet?
- Lines 347-348. Reference missing.
- How mood changes were assessed? Did the patient undergo a psychological consultation? Were mood questionnaires used to assess mood changes and alterations?
Minor comments
- Line 47-48. I suggest using the abbreviation “T2DM” for type 2 diabetes mellitus.
- Line 73 “are more likely adopt” should be replaced with “are more likely to adopt”
- Line 74 “dietary and activity” should be replaced with “diet and physical activity”
- Line 98 “a tendency for a drastic shift” should be replaced with “a tendency to shift”
- The authors should carefully check the text for some not defined abbreviations (i.e. lines 121-122 PAX, HMG20A)
- Line 186 “complain from” should be replaced with “complain of”.
- Line 195 “withing normal range” should be replaced with “within the normal range”.
- Lines 199-200 “it was decided that 199 she will be hospitalized for a short while until her glucose level is controlled”. Please, revise English language as appropriate.
- Line 206 “This plan did the trick for a while”: informal language. Please, revise.
- Line 257 “Waste circumference” should be replaced with “waist circumference”.
- Line 283 “anti-glycemic”. I suggest replacing with the word “anti-hyperglycemic” .
- Line 283 “a 10-years follow longitudinal Japanese”. Please, revise.
Author Response
Manuscript ID: ijerph-992042
Title: Intermittent fasting, dietary modifications, and exercise for the control of gestational diabetes and maternal mood dysregulation: a brief review and a case report
Response to Comments of Reviewer 1
We appreciate Reviewer 1’s helpful comments and concern for clarity as indicated by the provided comments. The comments are addressed line-by-line as shown below. Replies come underneath in red.
This short review and case report is focused on the possible beneficial effects of intermittent fasting on metabolic and affective alterations that occur in pregnancy complicated by GDM. Overall, the topic is interesting, as the first therapeutic approaches in GDM is medical nutritional therapy and novel dietary patterns are needed to achieve good glycemic control and prevent GDM complications. Furthermore, nowadays there is growing interest in fasting practice for the control of metabolic and other diseases. However, I believe that this work might be considered for publication only after an extensive revision. Specifically, the authors should address the following issues:
Major comments
- The first part of this review (introduction section) is not well organized. More specifically, information and evidence about different subjects (i.e. GDM overview, role of gut microbiome in the pathogenesis of GDM and GDM complications, mood dysfunction in GDM) are reported in a single paragraph in a random order. The authors should make an effort to reorganize this section following a more precise scheme. More in details, I suggest creating a paragraph (introduction) including lines 41-58 + lines 135-161 (overview of GDM epidemiology, pathophysiology, complications, therapy etc); then a paragraph on the role of gut microbiome in GDM pathophysiology etc.; a paragraph focusing on mood disorders and GDM etc.; and, lastly, a paragraph focusing on fasting practice.
Thank you very much; this comment is really helpful. Following the scheme illustrated by the reviewer along with similar ones offered by Reviewer 3, we altered the introduction and added new sections to address the key issues addressed in the title of this manuscript.
- In the introduction section, besides the epidemiologic data, the authors should mention also the main complications of GDM (Acta Diabetol. 2018 Dec;55(12):1261-1273. doi: 10.1007/s00592-018-1208-x; J Endocrinol Invest. 2018 Jun;41(6):671-676. doi: 10.1007/s40618-017-0791-y).
Yes, we elaborated on maternal and child-related complications of GDM (line 47).
- Line 54. The authors should briefly mention the role of epigenetic and environmental factors in the development of GDM and GDM complications (Int J Mol Sci. 2020 Jun 4;21(11):4020. doi: 10.3390/ijms21114020; Nutrients. 2020 Feb 19;12(2):525. doi: 10.3390/nu12020525).
Yes, we briefly mentioned the role of epigenetic and environmental factors in the development of GDM and GDM complications (line 166 and 216).
- Lines 114-117. It is worth reporting that during pregnancy not only adipose tissue but also gestational tissues contribute to the increase of pro-inflammatory cytokines and other molecules, which might favour the onset of GDM and the development of GDM complications (Acta Diabetol. 2019 Jun;56(6):681-689. doi: 10.1007/s00592-019-01304-x).
Yes, we elaborated on the contribution of gestational tissues to GDM pathology (line 202).
- Lines 149-162. In this section, the authors discuss the main dietary approaches in GDM. Certain dietary patterns (LGI diet, ketogenic diet) are reported as commonly adopted for the control of GDM. However, as yet, there is no consensus on which specific nutritional approach should be used in terms of total energy intake and macronutrient distribution. Moreover, the adoption of a ketogenic dietary approach is, to date, not recommended in pregnancy due to the lack of safety clinical data. The authors should clarify this concept, underlying that, overall, caloric restriction is not recommended during pregnancy, even in GDM, because it may have adverse effects on fetal birth weight. As for LGI diet, the authors should add that a considerable body of research suggests that the LGI diet is reasonably safe and might have positive effects on certain metabolic outcomes in GDM. By contrast, the LGI diet has not shown to have a marked influence on other obstetric maternal or fetal outcomes (body weight gain, birth weight, proportion of LGA, and macrosomia) so far (Nutrients. 2019 Jul 9;11(7):1549. doi: 10.3390/nu11071549).
Yes, we added a paragraph reporting on possible benefits and adverse effects of varying degrees of caloric restriction during pregnancy (line 329).
We also emphasized the safety of low GI diet during GDM. We spotted a meta-analysis that reports positive effects of low GI diet on fetal birth weight (line 303-307)
- In the case report section, relevant clinical data are missing or incomplete. Specifically, it is important to report pre-pregnancy BMI, blood glucose values at OGTT, fetal ultrasound parameters (head circumference, waist circumference, fetal weight) and, more importantly, the percentiles of these parameters.
Yes, in this revision we reported on pre-pregnancy BMI (line 357), blood glucose values at OGTT (line 376), fetal ultrasound parameters (head circumference, waist circumference, fetal weight) with percentiles of these parameters (Table 1).
- Lines 190-192 and 196-199. I believe that it is not necessary to report the personal opinion of the doctor, nor what the doctor told the patient. A case report should give objective data and information.
Yes, thank you. Sentences involving the personal opinion of the doctor were removed.
- Lines 203-205 and lines 213-215. Details about the dietary plans should be given, including calories/day, percentages of macronutrients (carbohydrates, lipids, proteins), number of meals etc. In the same way, the description of the “fasting” dietary plan, which is the main point of this review, is too general and synthetic. I suggest going into details.
Yes, we reported on daily caloric intake (line 394), number of meals (line 374 and 395), and the fasting regimen (line 395). We regret that the percentages of macronutrients (carbohydrates, lipids, proteins) were not determined by the diabetes clinic. Since this report was obtained from the patient a bit latter after birth, we could not estimate these percentages.
- Lines 208-209 and lines 226-227. Please, specify the percentile of fetal weight at this stage. Objective parameters are needed. Percentiles of biometric parameters should be reported to make comparisons and evaluate the effects of this dietary plan.
Yes, we added a reference to Table 1 in the indicated location. This table shows fetal growth change illustrated by relevant percentiles (line 372 and 387).
- Line 207. Pre-prandial and post-prandial glycaemic targets should be indicated. Which targets have been employed to establish the degree of glycaemic control?
Only a post-prandial glycaemic target less than 120 mg/dl was indicated (line 384).
- Lines 224-228. There is no information about glycaemic control after intermittent fasting dietary plan: did the patient achieve the prefixed glycaemic targets? What about the biochemical parameters (blood glucose, kidney function, electrolytes, urine ketones etc) during fasting intermittent diet?
Yes, we reported on these parameters in Table 2. Unfortunately, these measures were obtained only once during the third trimester, and we noted that as a limitation (line 678).
- Lines 347-348. Reference missing.
Yes, the relevant reference was added.
- How mood changes were assessed? Did the patient undergo a psychological consultation? Were mood questionnaires used to assess mood changes and alterations?
The patient’s psychological status was not assessed by a specific psychological measure, and it is reported here based on the patient’s narration, which we included in this submission as supplementary material. Although these narratives contain some sleep disturbance, stress, and mood-related symptoms; they are not enough to confirm a certain diagnosis—which will be inaccurate and misleading. So, we just reported on them as mood dysregulation.
Minor comments
- Line 47-48. I suggest using the abbreviation “T2DM” for type 2 diabetes mellitus.
Yes, we used the abbreviation “T2DM” for type 2 diabetes mellitus
- Line 73 “are more likely adopt” should be replaced with “are more likely to adopt”
Yes, we added the missing preposition.
- Line 74 “dietary and activity” should be replaced with “diet and physical activity”
Yes, “dietary and activity” was replaced with “diet and physical activity”.
- Line 98 “a tendency for a drastic shift” should be replaced with “a tendency to shift”
- The authors should carefully check the text for some not defined abbreviations (i.e. lines 121-122 PAX, HMG20A)
Yes, PAX, HMG20A were defined.
- Line 186 “complain from” should be replaced with “complain of”.
Yes, “complain from” was replaced with “complain of”.
- Line 195 “withing normal range” should be replaced with “within the normal range”.
Yes, “withing normal range” was replaced with “within the normal range”.
- Lines 199-200 “it was decided that 199 she will be hospitalized for a short while until her glucose level is controlled”. Please, revise English language as appropriate.
Based on the reviewer’s comment number 7, this sentence was removed since it comprises unnecessary details.
- Line 206 “This plan did the trick for a while”: informal language. Please, revise.
Yes, we have rewritten this sentence (line 382).
- Line 257 “Waste circumference” should be replaced with “waist circumference”.
Yes, “Waste circumference” was replaced with “waist circumference”.
- Line 283 “anti-glycemic”. I suggest replacing with the word “anti-hyperglycemic”.
Yes, “anti-glycemic” was replaced with “anti-hyperglycemic”.
- Line 283 “a 10-years follow longitudinal Japanese”. Please, revise.
Yes, we have rewritten this sentence (line 511).
We hope that the comments were properly handled and that the revised version will be suitable for publication.
Best regards,
Corresponding author

Reviewer 2 Report
This is an interesting review article and case report by Ali and Kunugi. I have a few comments, particularly in regards to organization and references. In the introduction section, it would be easier to read if the section was organized with subheadings. Also, it is unclear if references are referring to animal models or clinical studies, so this verification is required when necessary.
Minor comments:
- Please report the x-axis for Figure 2
- Some editing is required to check for spelling and grammar. For example, it should be waist circumference, not waste.
- why were jelly beans prescribed?
- was the patient prescribed therapy for her anxiety? I would think this would play a larger part in improving her mood compared to diet.
- How was her mood assessed?
Author Response
Manuscript ID: ijerph-992042
Title: Intermittent fasting, dietary modifications, and exercise for the control of gestational diabetes and maternal mood dysregulation: a brief review and a case report
Response to Comments of Reviewer 2
First of all, we would like to thank reviewer 2 for such a nice and supportive comment. The comments are addressed line-by-line as shown below. Replies come underneath in red.
This is an interesting review article and case report by Ali and Kunugi. I have a few comments, particularly in regards to organization and references. In the introduction section, it would be easier to read if the section was organized with subheadings. Also, it is unclear if references are referring to animal models or clinical studies, so this verification is required when necessary.
Yes, as the reviewer indicated, we have added subheadings for a better organization.
We have also indicated whether studies included animal or human subjects by using terms like experimentally, experimental evidence, rodents, rats, mice when we reported results of animal studies. We used terms like pregnant women when referring to human subjects.
Minor comments:
- Please report the x-axis for Figure 2
- Some editing is required to check for spelling and grammar. For example, it should be waist circumference, not waste.
Response to comment 1 and comment 2: Yes, thank you very much. Based on these comments, we have edited the text and the figures to improve their quality.
- why were jelly beans prescribed?
- was the patient prescribed therapy for her anxiety? I would think this would play a larger part in improving her mood compared to diet.
- How was her mood assessed?
Response to comment 3, comment 4, and comment 5: We need to make sure that the case reported in this manuscript is of a self-management attempt not an intervention offered to the patient by health professionals. So, all foods described in the report are patient’s personal choice based on her internet search for healthy food for diabetics. The patient voluntarily wanted to share her experience and agreed to cooperate with us to produce this report, and data were obtained latter after childbirth. Therefore, no ethical approval was obtained. We regret that the patient’s psychological status was not assessed by a specific psychological measure, and it is reported here based on the patient’s narration, which we included in this submission as a supplementary material. Although these narratives contain some sleep disturbance, stress, and mood-related symptoms; they are not enough to confirm a certain diagnosis—which will be inaccurate and misleading. So, we just reported on them as mood dysregulation. The patient did not receive any treatment for her psychological problems. We have clarified these aspects in the manuscript.
We hope that the comments were properly handled and that the revised version will be suitable for publication.
Best regards,
Corresponding author

Reviewer 3 Report
- Line 173: I believe that “fasting” should be replaced intermittent fasting.
- Line 183: I think that “This is her third pregnancy..” should be replaced by This was her third pregnancy…”.
- “Accordingly, she consumed 2 meals a day at a pace of 13-15 hours with complete lack of consumption of foods and drinks during this time. However, she drank a lot plain fluids and had light snacks during non-fasting hours.” These two sentences should be re-written to make them more clear for the readers.
- Line 296: Can authors provide more information about W-containing peptides?
- Figure 1. Can Authors explain why graphs b (abdominal circumference), and c (height of the uterus) start from 0? It is unclear. Moreover, the Authors should indicate whet the patients start with intermittent fasting.
- Figure 2. X-axis is not descripted.
- The delivery data should be added.
- Did the human study (case study) receive ethical approval?
Author Response
Manuscript ID: ijerph-992042
Title: Intermittent fasting, dietary modifications, and exercise for the control of gestational diabetes and maternal mood dysregulation: a brief review and a case report
Response to Comments of Reviewer 3
First of all, we thank the reviewer for such specific and enlightening comments. The comments are addressed line-by-line as shown below. Replies come underneath in red.
- Line 173: I believe that “fasting” should be replaced intermittent fasting.
Yes, we replaced “fasting” with “intermittent fasting” in all instances referring to fasting as a therapeutic modality.
- Line 183: I think that “This is her third pregnancy..” should be replaced by This was her third pregnancy…”.
Yes, we replaced “This is her third pregnancy..” by “This was her third pregnancy…” (line 358).
- “Accordingly, she consumed 2 meals a day at a pace of 13-15 hours with complete lack of consumption of foods and drinks during this time. However, she drank a lot plain fluids and had light snacks during non-fasting hours.” These two sentences should be re-written to make them more clear for the readers.
Yes, we have rephrased this part of the text to make it clearer (line 395).
- Line 296: Can authors provide more information about W-containing peptides?
Yes, we briefly elaborated on W residues and their health benefits (line 524).
- Figure 1. Can Authors explain why graphs b (abdominal circumference), and c (height of the uterus) start from 0? It is unclear. Moreover, the Authors should indicate whet the patients start with intermittent fasting.
Thank you very much for this comment. Data elements for the 13th week and the 16th weeks were not documented in the patient’s record and were left blank. These entries were removed, and graph b and c were modified.
We also created Figure 4 to illustrate on the timeline of application of dietary modification and intermittent fasting.
- Figure 2. X-axis is not descripted.
Yes, we modified this Figure.
- The delivery data should be added.
Yes, we added some details of childbirth (line 411).
- Did the human study (case study) receive ethical approval?
We need to make sure that the case reported in this manuscript is of a self-management attempt not an intervention offered to the patient. The patient voluntarily wanted to share her experience and agreed to cooperate with us to produce this report. Therefore, no ethical approval was obtained. However, we follow research ethics while presenting patient’s data e.g., patient’s name, ID number, and hospital name on the ultrasound images were masked.
We hope that the comments were properly handled and that the revised version will be suitable for publication.
Best regards,
Corresponding author

Round 2
Reviewer 1 Report
The authors successfully addressed the raised issue.
Author Response
Manuscript ID: ijerph-992042
Title: Intermittent fasting, dietary modifications, and exercise for the control of gestational diabetes and maternal mood dysregulation: a review and a case report.
Response to Comments of Reviewer 1
We would like to express our gratitude to the reviewer for the constructive comments that helped improve this work.
Best regards,
Corresponding author
Reviewer 3 Report
I have no more comments to the Authors.
Author Response
Manuscript ID: ijerph-992042
Title: Intermittent fasting, dietary modifications, and exercise for the control of gestational diabetes and maternal mood dysregulation: a review and a case report.
Response to Comments of Reviewer 3
We would like to express our gratitude to the reviewer for the constructive comments that helped improve this work.
Best regards,
Corresponding author